# Inflammation Regulates the Multi-Step Process of Retinal Regeneration in Zebrafish

**DOI:** 10.3390/cells10040783

**Published:** 2021-04-01

**Authors:** Mikiko Nagashima, Peter F. Hitchcock

**Affiliations:** Department of Ophthalmology and Visual Sciences, University of Michigan, Ann Arbor, MI 48105, USA; mikiko@umich.edu

**Keywords:** Müller glia, microglia, photoreceptor, cytokine, damage, reprogramming, proliferation

## Abstract

The ability to regenerate tissues varies between species and between tissues within a species. Mammals have a limited ability to regenerate tissues, whereas zebrafish possess the ability to regenerate almost all tissues and organs, including fin, heart, kidney, brain, and retina. In the zebrafish brain, injury and cell death activate complex signaling networks that stimulate radial glia to reprogram into neural stem-like cells that repair the injury. In the retina, a popular model for investigating neuronal regeneration, Müller glia, radial glia unique to the retina, reprogram into stem-like cells and undergo a single asymmetric division to generate multi-potent retinal progenitors. Müller glia-derived progenitors then divide rapidly, numerically matching the magnitude of the cell death, and differentiate into the ablated neurons. Emerging evidence reveals that inflammation plays an essential role in this multi-step process of retinal regeneration. This review summarizes the current knowledge of the inflammatory events during retinal regeneration and highlights the mechanisms whereby inflammatory molecules regulate the quiescence and division of Müller glia, the proliferation of Müller glia-derived progenitors and the survival of regenerated neurons.

## 1. Inflammation and Tissue Regeneration

Inflammation is the tissue-based response to invasive pathogens or cellular injury. In injured tissues, factors released from dying cells initiate activation and recruitment of both resident and circulating immune cells and promote the synthesis and release of soluble factors, cytokines and chemokines. Rapid, acute-inflammation, mediated by the innate immune system, functions to restore tissue homeostasis and promote healing. Increasing evidence also indicates that acute inflammation is an essential regulator of tissue regeneration. For example, in tissues that rely on cellular reprogramming to regenerate, the activation of inflammatory signaling pathways enhances chromatin remodeling and the accessibility of reprogramming factors to the DNA [1,2]. In regenerating tissues, inflammation also promotes proliferation among the reprogrammed cells and their progeny [3,4,5,6]. In contrast to acute inflammation, dysregulation of inflammation in response to cell death can result in chronic inflammation, which is both detrimental to the injured tissue and incompatible with regeneration [7]. In zebrafish, acute inflammation is required for successful regeneration of multiple tissues, including fin, heart and central nervous system (see recent review by Iribarne, 2021 [8]). Resolving inflammation once regeneration is complete is essential and equally complex. Importantly, a comprehensive understanding of the role and regulation of inflammation during tissue regeneration may lead to novel cell-based replacement therapies in humans based on features of the immune response.

In injured tissues, inflammation is initiated by activation of pattern recognition receptors (PRRs) by damage-associated molecular pattern (DAMPs) molecules, including the chromatin-associated nuclear protein high mobility group 1, heat shock proteins, purine metabolites, such as adenosine triphosphate (ATP), and uric acid. In healthy tissues, endogenous DAMPs are sequestered intracellularly. Following injury, cells expose or release DAMPs into the extracellular space, activating classic PRRs, such as Toll-like receptors, nucleotide-binding oligomerization domain (NOD)-like receptor, and retinoic acid-inducible gene (RIG)-I-like receptors, and non-canonical PRR, such as receptor for advanced glycation end-products and G-protein-coupled receptors [9,10,11,12]. Downstream signaling of PRRs leads to the activation of nuclear factor kappa Β (NF-κΒ) transcription factors and the mitogen-activated protein kinases (MAPKs), p38a and cJun N-terminal kinase (JNK), and the synthesis of interferon-α [13,14]. NF-κΒ drives the expression of a large number of pro-inflammatory genes [15]. Similarly, activation of p38a and JNK leads to the activation of transcription factors, cAMP response element-binding protein (CREB), CCAAT enhancer-binding protein beta (C/EBPβ) and activator protein 1 (AP1) and induction of pro-inflammatory mediators [16,17]. Such mediators include adhesion molecules and extracellular matrix proteins, which play roles in recruitment of immune cells to the injury, and cytokines, a diverse group of small, soluble factors that activate, amplify, and terminate inflammation. Various cell types, including immune-cells, release cytokines. Based on the nature of response, cytokines can be classified into lymphocyte growth factors, pro-inflammatory factors, anti-inflammatory factors, and factors that polarize the immune response to antigens [18]. Pro-inflammatory cytokines, such as Interleukin1β (IL1β), IL6, and tumor necrosis factor alpha (TNFα), are predominantly expressed by immune cells and are involved in facilitating acute inflammation [19]. Anti-inflammatory cytokines, such as IL4, IL10, IL11, IL13, and transforming growth factor beta (TGFβ), have immunomodulatory functions and act by repressing pro-inflammatory signaling pathways and upregulating the production of anti-inflammatory cytokines. [19]. Cytokines act through their specific receptors expressed on the surface of target cells. Depending on the cell type that the receptor is expressed, cytokines can act on themselves (autocrine), nearby cells (paracrine), and distant cells (endocrine). A given cytokine receptor can be restricted to a single cell type or present on the surface on many cell types, therefore, cytokines have context-dependent roles that dictate their physiological functions [20]. Upon binding to its receptor, cytokines activate complex intracellular signaling cascades that regulate the activation and induction of downstream genes. Binding of the pro-inflammatory cytokines, IL1β and TNFα, to its receptors, Il1 receptor and TNFα receptor I, respectively, further activate NF-κΒ and MAPKs [21,22]. This positive feedback loop serves to amplify the inflammatory response. The Janus kinase/ Signal transducer and activator of transcription proteins (JAK/STAT) pathway is a pleiotropic major signaling pathway activated by a wide range of cytokines, including members of the interleukin family [23,24,25]. Upon binding to its receptor, cytokines activate the associated JAK kinases, resulting in phosphorylation of STAT family proteins [26]. Numerous cytokines and other ligands can activate the receptor-associated JAKs [24]. JAK-STAT signaling interacts synergistically or antagonistically with other cascades, such as MAPKs and phosphoinositide 3 kinase (PI3K) [23]. The complexity of cytokine action relies on a cell’s ability to activate and inactivate complex arrays of interacting signaling pathways [27].

In this review, we summarize the current literature regarding the cellular and molecular components of inflammation and how these components govern neuronal regeneration in the retina of zebrafish. First, we provide a brief overview of persistent and regenerative neurogenesis in the zebrafish retina, comparing and contrasting growth with regeneration. Second, we review the activation of microglia, the intrinsic macrophages of the brain and retina and the key cellular players in the inflammatory response. Third, we highlight recent evidence showing that inflammation reprograms Müller glia, which function as an intrinsic stem cell in the retina, and how acute inflammation drives proliferation among Müller glia and Müller glia-derived progenitors. We propose a hypothesis whereby the magnitude and duration of the acute inflammatory response functions to quantitatively match the amplification of Müller glia-derived progenitors to the extent of cell death. Fourth, we discuss possible mechanisms that govern the resolution of the acute inflammatory response during neuronal regeneration and describe data showing how alterations in the resolution phase compromise the health and survival of regenerated neurons. Finally, we provide a summary and note directions for future research.

## 2. Persistent and Regenerative Neurogenesis in Adult Zebrafish

In the retina (and brain) of adult teleost fish there are two forms of neurogenesis, persistent neurogenesis, which is life-long and associated with growth, and regenerative neurogenesis, which is acutely activated by cell death [28,29,30]. In the retina, persistent neurogenesis originates from two spatially-separate stem cell niches and lineages. The ciliary marginal zone is appended to the anterior edge of the retina and contains neuroepithelial stem and progenitor cells that generate all types of retinal neurons, with the exception of rod photoreceptors. The ciliary marginal zone adds annuli of new retinal cells as the eye and animal grow and contributes to the striking postembryonic growth of the retina [29,31,32,33]. In contrast, rod photoreceptors originate from Müller glia [34], which in the uninjured retina divide occasionally and asynchronously to produce fate-restricted rod precursors that migrate to the layer of photoreceptor nuclei [35,36]. There, rod precursors also divide and differentiate exclusively into rod photoreceptors that are insinuated into the extant lawn of both rod and cone photoreceptors. The mechanisms that determine when an individual rod photoreceptor is differentiated and what constrains these cells to a rod photoreceptor fate, however, are not understood. This growth-associated neurogenesis is regulated by the growth-hormone-insulin-like growth factor 1 (IGF-1) axis [37], the mammalian-like growth hormone system that ties the growth of various tissues to the indeterminant growth of the fish.

Identified as a population of dividing cells that reside within the parenchyma of the teleost retina, rod precursors were originally proposed as the cellular origin of regenerating neurons [38]. After nearly two decades of studies using antibody markers of proliferation and techniques for performing lineage tracing, Müller glia were definitively identified as the cellular origin of regenerated neurons (Figure 1A) [34,39,40,41].

In contrast to the occasional and asynchronous cell division during retinal growth, following retinal injury and cell death, Müller glia enter the cell cycle in a highly synchronized manner [42]. Following this, the progeny of Müller glia proliferate rapidly, migrate to the various cellular layers of the retina that have been damaged, and there reveal their multipotential nature by differentiating into any type of retinal neuron [43]. In the teleost retina, the parallels between the growth-associated rod genesis and neuronal regeneration are obvious, but, as we review below, neuronal regeneration originating from Müller glial lineages is driven by the inflammatory response [6,44]. Perhaps inferred *a priori*, it was recently established that the cell fate-determination programs that regulate retinal development are recapitulated during the final stages of regeneration and serve to govern differentiation, process of outgrowth and the re-establishment of synaptic circuits among the regenerated neurons [45].

Several experimental paradigms using zebrafish have been developed to kill retinal neurons: (1) photolytic lesions, which selectively kill photoreceptors [34,46]; (2) inducible genetically-targeted ablation via bacterial nitroreductase systems [6], which can target any cell type; (3) puncture wounds that penetrate the globe and retina [47] which elicits local regeneration at the site of the puncture; and 4) intraocular injections of the sodium-potassium ATPase inhibitor, ouabain [42,48,49], which causes variable and unselective cell death that is dose dependent. The retinas of teleost fish are avascular. Photolytic lesions and nitroreductase ablations do not damage blood vessels surrounding the retina, whereas puncture wounds and ouabain injections cause injury to the vessels supplying nutrients to the globe and/or retina. In injuries to the forebrain of zebrafish, peripheral macrophages appear at the injury site prior to activated microglia [50], indicating that circulating and resident immune cells respond over different time-frames and, therefore, may have separate functions. It remains to be determined if the mechanisms that initiate and regulate retinal regeneration differ between paradigms that injure blood vessels, and allow peripheral immune cells to invade the retina, and paradigms that do not damage ocular and retinal vasculature and where regeneration is mediated only by cells intrinsic to the retina [51].

## 3. Activation of Microglia

Microglia are the intrinsic macrophages of the central nervous system. In the adult central nervous system under homeostatic conditions, these cells have a ramified morphology and function to actively surveil brain parenchyma. In the retina, microglia reside at the interface of the nuclear and synaptic layers, where they likely also play a role in maintaining synaptic integrity [52,53]. Following cell death, microglia in the zebrafish retina undergo rapid and marked changes in morphology, from ramified to ameboid, and migrate to sites of cell death [42,49,54]. Along with changes in morphology, there is also a rapid induction in the expression of genes encoding inflammatory cytokines [44,54,55]. The essential role of microglia during retinal regeneration in zebrafish has been demonstrated by studies that utilize genetic and pharmacological techniques to ablate microglia. Depleting microglia dampens proliferation among Müller glia [6,44,51]. Although the known slate of molecules released by the microglia is incomplete, analysis of RNAseq from isolated microglia/macrophages and single-cell RNAseq identified microglia-specific genes upregulated following a retinal injury that encode inflammatory factors, such as *il-6 subfamily cytokine m17*, *granulin-1*, -*2*, and -*a*, *tnfβ*, *interleukin 1β,* and *lectin* [55,56].

An equally important function of activated microglia is to engulf apoptotic cells [49]. This function is initiated by a complex network of “Find-me” and “Eat-me” signals expressed by the apoptotic cells and the specialized phagocytic receptors and bridging molecules present on microglia [57,58,59]. During the early stage of apoptosis, dying cells release extracellular “Find-me” signals, such as lysophosphatidylcholine (LPC) and nucleotides, that facilitate the recruitment of phagocytes. “Eat-me” signals, including phosphatidylserine and intercellular adhesion molecule exposed on the surface of apoptotic cells, are recognized by their receptors expressed on the phagocytes. Upon recognition, phagocytes activate intracellular signaling cascades regulating rearrangement of cytoskeleton and the post engulfment processes of cargo engulfment, phagosome maturation, and cargo degradation [60]. Although molecular mechanisms that govern phagocytosis of microglia in the context of regenerating retinas is not well elucidated, recent transcriptome analysis also identified genes and pathways involved in phagocytosis of extracellular substrates [55]. It remains to be determined if regeneration-specific molecules or mechanisms are involved in clearing apoptotic cells [61].

## 4. Quiescence, Reprogramming and Proliferation in Müller Glia

Müller glia are specialized form of radial glia that are present in all vertebrate retinas. Their nuclei reside in the inner nuclear layer, and they possess radial processes that span the entire thickness of the retina and tangential processes that occupy all the extracellular space between neurons [62]. In a healthy retina, Müller glia are in a quiescent state and function to provide structural integrity and to support neuronal homeostasis [62]. Müller glia also display features of typical astrocytes, within both homeostatic and injury conditions [63]. Given their common role of maintaining homeostasis in all vertebrate retinas and their unique role in mediating retinal regeneration in fish, Müller glia are intensively studied in the retinas of all vertebrates.

In zebrafish there are several mechanisms that actively maintain Müller glia in a quiescent state and suppress reprogramming (Figure 1B). These include cell-cell contact and the local neurotransmitter milieu [64,65,66,67]. In addition, molecules that are components of cytokine networks regulate signaling pathways in an autocrine manner that maintain Müller glia in a quiescent state. Müller glia secrete Tgfβ3, which phosphorylates Smad3 and collaborates with Protein Phosphatase 2/PP2A to suppress the expression of reprogramming factors [68]. This non-canonical Tgfβ signaling pathway also positively regulates Notch signaling components, *hey1* and *Dil4*, which are required for quiescence [64,67,69,70]. Consistent with this, downregulating and deactivating Tgfβ signaling is necessary for Müller glia to reprogram and proliferate [71,72]. Molecular mechanisms that promote downregulation of Tgfβ are not known, though feedback via other cytokines may be involved.

Müller glia in all vertebrates are exquisitely sensitive to cellular injury within the retina, likely through changes in neurotransmitter dynamics, alterations in intercellular contacts or the presence of soluble inflammatory factors. In response to these signals, Müller glia mount a complex, multi-phase response. Initially, Müller glia undergo a transient reactive gliosis [73], characterized by upregulation of the intermediate filament protein, glial fibrillary acidic protein. This gliotic state rapidly transitions to reprogramming. Reprogrammed Müller glia down-regulate a subset of glia-specific genes, e.g., *rlbp1a* and *glul*, upregulate genes that encode reprogramming factors, e.g., *oct4,* and upregulate genes indicative of retinal stem and progenitor cells, e.g., *pax6a*, *sox2*, and *rx1* [42,74,75,76]. The rapid and dynamic reprogramming of Müller glia is followed closely by their entry into the cell cycle, which is characterized by interkinetic nuclear migration and a single asymmetric cell division that produces a Müller glia-derived progenitor and a post-mitotic Müller glia [42,77,78]. As they divide, Müller glia maintain their original morphology; progenitors are generated by nuclear division and somatic cytokinesis. Post-mitotic Müller glia then re-express glial markers and re-differentiate. The transient reprogramming of Müller glia can best be characterized as a partial and reversible dedifferentiation. Prior to reprogramming, Müller glia upregulate the expression of numerous cytokines and chemokines [54,55,56,79].

In response to pro-inflammatory molecules released by dying cells, microglia (see above) and Müller glia themselves, activated intracellular pathways promotes reprogramming and cell division (Figure 1C). Following their death, photoreceptors release Tnfα [80]. Using gain- and loss-of-function approaches, Nelson et al. determined that Tnfα was both necessary and sufficient to reprogram Müller glia, which was mediated by upregulating the expression of the reprogramming factor genes, *ascl1a* and *stat3*. This foundational study provided the first evidence that dying neurons are a source of inflammatory factors with the ability to reprogram Müller glia. Further, Western blot analysis revealed levels of the active, cleaved form of Tnfα are highest at the peak of photoreceptor death. The most parsimonious interpretation of these results is that cytokines and other factors released by dying cells signal directly to Müller glia, however, it cannot be excluded that microglia serve as an intermediary in this signaling cascade. Interestingly, once reprogramming has commenced, Müller glia also synthesize Tnfα, which functions to regulate the number of Müller glia-derived progenitors. In addition to Tnfα, Müller glia secrete the cytokines Leptin and the Il6 family of cytokines, which function to synergically activate the Jak/Stat3 pathway through the Leptin receptor and a signaling component, Gp130, respectively, which, in turn, drives cell division [81,82].

Following reprogramming, the fundamental feature of Müller glia is entry into and progression through the cell cycle. A recent study from our lab showed that the zebrafish ortholog of mammalian Midkine, Midkine-a, does not play a role in reprogramming Müller glia. Rather, it functions in an autocrine manner to regulate the progression of Müller glia through the cell cycle [83]. Midkine-a is a heparin-binding cytokine/growth factor that has multiple roles in neural development, tissue repair and disease [84]. In all vertebrate tissues studied to date, injury induces a rapid induction of *midkine,* and Midkine has become a *bona fide* molecular component of both acute and chronic inflammation [85]. In the zebrafish retina, cell death induces the rapid expression of Midkine-a in Müller glia. When *midkine-a* is genetically deleted, following the death of photoreceptors Müller glia undergo a normal reprogramming response, as evidenced by the increased expression of the reprogramming genes, *ascl1a*, *stat3*, and *lin28*, phosphorylation of Stat3 and entry into the G1 phase of the cell cycle. However, in the absence of Midkine-a, transition from G1 to S phase is blocked, and this results in significantly reduced proliferation and the selective failure to regenerate cone photoreceptors. (Although Müller glia fail to progress through the cell cycle, ablated rod photoreceptors are replenished from the extant pool of rod precursors.) Interestingly, Müller glia that fail to progress through the cell cycle undergo prolonged reactive gliosis, a pathological hallmark of Müller glia in the retinas of mammals. This study also determined that the putative Midkine-a receptor, anaplastic lymphoma kinase, is upstream of the helix-loop-helix (HLH) regulatory protein, Id2a, and the retinoblastoma gene, *p130*, which directly regulates cyclins and cyclin-dependent kinases. It was found that Midkine-a normally induces a brief, transient upregulation of Id2a, which suppresses cell cycle inhibitor p130, thereby allowing Müller glia to enter both S and the subsequent phases of the cell cycle. This study demonstrated that in the zebrafish Müller glia, Midkine-a functions as a component of the inflammatory response to cell death, functioning as a core component of the mechanisms that regulate progression through the cell cycle. Insights into the mechanisms that govern cell division in Müller glia highlight a gap in our knowledge regarding the mechanisms that limit Müller glia to a single round of cell division. Unregulated cell division is a concern that accompanies all potential therapies that use stem or progenitor cells to replace ablated retinal neurons. It will be important also to identify the molecular mechanisms that constrain proliferation of Müller glia and their subsequent rapid return to quiescent state. It is possible that factors that promote glial quiescence in mice may play a similar role in the Müller glia of zebrafish. In mice following retinal injury, the transient downregulation of the nuclear factor I factors, Nfia, Nfib, and Nfix, is followed by their significant upregulation [56]. Genetic depletion of the NFIs promotes the expression of genes associated with cell cycle regulation and neurogenesis and induces proliferation of Müller glia, indicating that in mice the NFIs maintain Müller glia in a quiescent state.

## 5. Inflammation and the Amplification of Müller Glia-Derived Progenitors

In the regenerating retina of zebrafish, Müller glia and the Müller glia-derived progenitors are distinct populations of dividing cells. After their asymmetric division, which requires acute inflammation, maternal Müller glia re-differentiate into a quiescent state, whereas Müller glia-derived progenitors, transit amplifying cells that are the immediate antecedents of regenerated neurons, undergo rapid proliferation [42,77]. Given that proliferation of Müller glia and their progeny closely overlap in time and space, it can be challenging to uncouple molecular functions between these two populations of cells. Nonetheless, recent studies have revealed links between inflammation and proliferation of Müller glia-derived progenitors (Figure 2C). In an injured retina, inflammation and microglia-derived factors activate mTOR signaling in Müller glia-derived progenitors [86]. Pharmacological suppression of mTOR signaling, after the single round of division among Müller glia, reduces the number of Müller glia-derived progenitors [86]. Importantly, the effect of mTOR inhibition on proliferation is transient. Once mTOR suppression is withdrawn, Müller glia-derived progenitors regain the ability to progress through the cell cycle [86]. Similarly, depleting microglia impairs the proliferation of Müller glia-derived progenitors, and cessation of the drug administration results in repopulation of microglia, onset of proliferation and neuronal regeneration [51]. Although this study doesn’t exclude the possibility of compensatory proliferation of rod precursors, these results clearly demonstrate that modulating inflammatory components of the injury response alters proliferation among Müller glia-derived progenitors. Finally, a recent study from our lab showed that suppressing the inflammatory response with the glucocorticosteroid, dexamethasone, significantly diminishes proliferation of Müller glia and Müller glia-derived progenitors (Figure 2C; Silva et al. 2020 [44] and see below). It should be noted, however, that immune suppression during retinal regeneration may give rise to context- or cell-type dependent results. Altering the timing of immune suppression can either inhibit or accelerate the selective regeneration of rod photoreceptors [6].

The studies summarized above show that suppressing inflammation suppresses the proliferation of Müller glia-derive progenitors. Complementing these results, a recent study from our lab demonstrated that enhancing the acute inflammatory response induces hyperproliferation among these cells [44]. This study investigated the function of the inflammatory metalloproteinase-9 (Mmp-9) during the regeneration of photoreceptors. Mmp-9 is a secreted gelatinase, first purified from neutrophils and monocytes [87,88]. It is well established that inflammatory cytokines can induce the expression of Mmp-9, and, in turn, Mmp-9 can cleave cytokine precursors and mature cytokines into activate and inactivate forms [89]. Our study showed that *mmp-9* is strongly upregulated *de novo* in reprogrammed Müller glia and Müller glia-derived progenitors. Notably, in *mmp-9* mutants, a subset of inflammatory cytokines, including *tnfα*, were elevated, indicating that Mmp-9 likely plays a role in resolving the acute inflammatory response (see below). We found, too, that the absence of *mmp-9* results in the over production of Müller-glia derived progenitors and, interestingly, an overproduction of regenerated photoreceptors. These results were interpreted to show that Mmp-9 negatively regulates proliferation of Müller glia-derived progenitors, likely facilitating resolution of the inflammatory response (see below). The absence of Mmp-9 results in increased levels of inflammatory molecules, which were sufficient to stimulate the excess proliferation (Figure 2C).

A well-known observation, but one that is understudied, is the positive correlation between the extent of cell death in the retina and the degree of proliferation among the Müller glia-derived progenitors [90,91]. The amplification of Müller glia-derived progenitors is the event that quantitatively matches the number of regenerated neurons to the extent of cell death. An unanswered question is what links the amplification of these neural progenitors to the number of dying neurons? Based on our studies and those cited above, we hypothesize that the acute inflammatory response serves first to reprogram Müller glia and, second, to regulate proliferation, which serves to match the number of Müller glia-derived progenitors to the number of dying cells. This hypothesis implies that the sum of the factors released by dying cells determines the number of microglia that are activated and the number of Müller glia that are reprogrammed. Together the concentration and/or perdurance of inflammatory molecules released into the extracellular environment by microglia and Müller glia regulate cell cycle kinetics and or timing of cell cycle exit of the Müller glia-derived progenitors, thereby determining the number that are produced. In this manner, the extent of cell death is translated into the magnitude of the proliferative response and the number of regenerated neurons. The flexible amplification of the Müller glia-derived progenitors is the only point in the multistep process of neuronal regeneration that is sufficient to create a pool of neural progenitors that is equal to the number of ablated neurons. A future direction of research, therefore, will be to identify the components of the acute inflammatory response that control cell cycle kinetics and the timing of cell cycle exit among Müller glia-derived progenitors.

## 6. The Resolution of Inflammation

Resolving acute inflammation is not a passive or simple process [7,92]. Similar to the acute inflammatory phase, complex, tightly regulated factors and signaling pathways control the resolution of acute inflammation [93,94]. This is an important area of research, because it is well-established that uncontrolled or unresolved inflammation can lead to chronic inflammation, tissue damage and inflammatory disease [95]. The proper resolution of inflammation is a critical element of retinal regeneration and must involve eliminating and/or downregulating injury cues originating from all cellular sources and removing pro-inflammatory factors from the extracellular milieu. Our recent study is illustrative of the consequences of the dysregulation of injury-induced inflammation. In the absence of Mmp9, retinas exhibited elevated levels of a subset of inflammatory cytokines and the persistent activation of microglia. We discovered that in Mmp-9 mutants the survival and maturation of regenerated cone photoreceptors was compromised [44]. We inferred from these observations that Mmp9 likely catalyzes active inflammatory cytokines and/or receptors to resolve the acute inflammation and the resulting persistence of the inflammation was toxic to regenerated cone photoreceptors (Figure 2C). Interestingly, if in Mmp-9 mutants this post-regeneration inflammation was pharmacologically suppressed, the selective death of regenerated cone photoreceptors was rescued, indicating that the resolution of inflammation is critical to reestablish a favorable environment for the survival of regenerated neurons. Although in this instance suppressing inflammation was sufficient to rescue the death of cone photoreceptors, the simple catabolism of inflammatory cytokines alone is not sufficient to resolve inflammation [7]. Actively eliminating the cellular sources of pro-inflammatory signals is also a necessary component of inflammatory resolution. This relies upon, for example, phagocytosis of apoptotic cells by microglia and shifting gene expression in microglia cells away from pro-inflammatory and to anti-inflammatory programs [96,97,98,99,100]. We infer that in the zebrafish, phagocytosis by microglia induces an anti-inflammatory phenotype, and this serves, in part, to increase levels of anti-inflammatory molecules and resolve the acute inflammation induced by the original injury. Given that the magnitude and/or duration of an acute inflammatory response will doubtless affect many aspects of retinal regeneration, it is essential to seek a mechanistic understanding of both the induction and resolution of this complex process.

## 7. Summary and Future Directions

The overarching goal of studying retinal regeneration in zebrafish is to gain insights into potential therapeutic approaches for treating or curing neurodegenerative diseases in the human retina and brain. Inflammation is a universal response to cell death and underlies many biological processes, including cellular reprogramming, proliferation and cell survival. In zebrafish, Müller glia integrate acute inflammatory signals and activate signaling pathways that reprogram these cells into a stem cell state and promote entry into the cell cycle. In addition, inflammatory signals drive the progression of Müller glia-derived progenitors through the cell cycle. If aspects of acute inflammation persist, this compromises the survival and maturation regenerating neurons. Although evidence is clear that in zebrafish inflammation governs regenerative neurogenesis, much less is known about the molecular mechanisms regulate resolution of the acute inflammatory response.

A fundamental insight into the molecular biology of Müller glia in mammals was the discovery that in adult retinas these cells share a common gene expression profile with late stage retinal progenitors [101]. This led to the suggestion that Müller glia may be a form of late-stage retinal progenitor with a latent ability to generate neurons [102]. However, cell death in the mammalian retina leads to a strikingly different response. Müller glia respond to injury by a marked gliotic response, which is thought to be neuroprotective, as evidenced by the release of trophic factors that promote neuronal survival [62]. However, dysregulation of the supporting function of Müller glia is detrimental. In mice the forced ectopic expression of the bHLH transcription factor, Ascl1, when combined with cell death is sufficient to stimulate a Müller glia to enter the cell cycle [103]. The efficiency of this response is enhanced when the expression of Ascl1 and retinal injury are combined with a treatment of the histone deacetylase inhibitor, trichostatin-A (TSA), indicating that chromosome structure in mammalian Müller glia may limit the regenerative capacity of these cells [104,105]. These results point toward common molecular mechanisms for reprogramming Müller glia in all vertebrates. Interestingly, however, inflammation may play a fundamentally different role in the induced retinal regeneration in mice. A recent study revealed that during the TSA-mediated retinal regeneration, the ablation of microglia enhances the neurogenic capacity of Müller glia [106]. This suggests that in mammals, suppressing the acute inflammatory response improves regeneration, which contrasts with results observed following similar experimental manipulations in zebrafish (and post-hatch chick; see Gallina et al. 2014 [107]). The abnormal activation of microglia following ablation of TGFβ receptor 2 induces secondary gliosis of Müller glia, suggesting that in mice microglia-mediated inflammation exacerbates the gliotic response of Müller glia [108]. Whether or not the acute inflammatory response in the retinas of mammals and zebrafish plays fundamentally different roles in reprogramming Müller glia is a topic that should be investigated further. In addition, it should be considered that the molecular nature of the inflammatory signals may be context and species dependent. A recent study using the chick demonstrated that microglia-derived factors stimulate NF-κβ signaling in Müller glia, and this initiates the activation and reprogramming of Müller glia [109]. Importantly, the subsequent proliferation of Müller glia-derived progenitors requires suppression of NF-κΒ signaling, suggesting dual regulation of inflammation in reprogramming and proliferation of Müller glia. In the chick retina, inflammation activates reprogramming factors, however, persistent activation of inflammatory signaling negatively regulates proliferation of Müller glia-derived progenitors [109]. Although the role of NF-κΒ signaling in zebrafish is not well elucidated, studies demonstrate that epigenetic and transcriptional components of NF-κΒ signaling are differentially regulated in fish and mammals [56,79]. Similarly, another avenue for future research is to identify the mechanisms that regulate the amplification of Müller glia-derived progenitors. Although Müller glia can be reprogrammed, in mammals the proliferation of Müller glia-derived progenitors is slight, at best, and fails to match the extent of cell death. Understanding the mechanisms that regulate the proliferation of these cells in zebrafish may improve the efficiency of the regenerative response in mammals and help develop immune system-based approaches for successfully producing retinal (and central nerves system) regeneration in humans.

Finally, in zebrafish cells intrinsic to the retina are sufficient to activate and regulate retinal regeneration. Nonetheless, a study investigating the involvement of circulating regulatory T-like cells in retinal regeneration suggests involvement of circulating cells in retinal regeneration [110]. Regulatory T cells are a specialized subtype of T cells that maintain tolerance to self-antigens through an immunomodulatory function that restrains excessive inflammatory responses to infection or tissue damage [111,112]. Following a stab wound that penetrates both the globe and retina, forkhead box protein P3a (Foxp3a)-positive, zebrafish regulatory T-like cells (zTreg cells) infiltrate the retina at around 4 days post injury [110]. This delayed infiltration may imply that an acute response of the innate immune systems plays a role in activating and recruiting elements of the adaptive immune system. When zTreg cells were genetically ablated post-injury, there was a significant reduction in the number of proliferating cells observed at 7 days post lesion. Based on the timing of when the zTreg cells enter the retina and when diminished proliferation is observed following their ablation, the zTreg cells are likely regulating proliferation among Müller glia-derived progenitors. It is unknown if the zTreg cells function to reprogram Müller glia and/or stimulate their entry into the cell cycle. Regardless, these results point to a potential role for blood-borne cells during neuronal regeneration in vertebrate retinas (see Pesaresi et al. 2018 for an intriguing example of this [113]) and, as noted above, emphasizes the need to carefully evaluate results based on injury paradigms. Photolytic lesion induces apoptotic cell death, whereas cell death induced by the sodium-potassium ATPase inhibitor, ouabain, consists of concurrent components of apoptotic and necrotic pathways [114,115]. It should be noted that different pathways leading to cell death induce different cytokines and DAMPs, and, thereby, may activate distinct inflammatory cascades [116]. Additional studies need to be performed to parse the roles of intrinsic and circulating cells during retinal regeneration in zebrafish and to determine the potential roles these cells play in the experimental models of retinal regeneration in mammals. Finally, it should be noted that regulation of inflammation extends beyond cytokines. The complement cascade and chemical mediators such as leukotrienes and histamines play roles in regulating acute and chronic inflammation [117,118,119]. Future comprehensive studies will provide further understanding of neuroinflammation and tissue regeneration.

## Figures and Tables

**Figure 1 cells-10-00783-f001:**
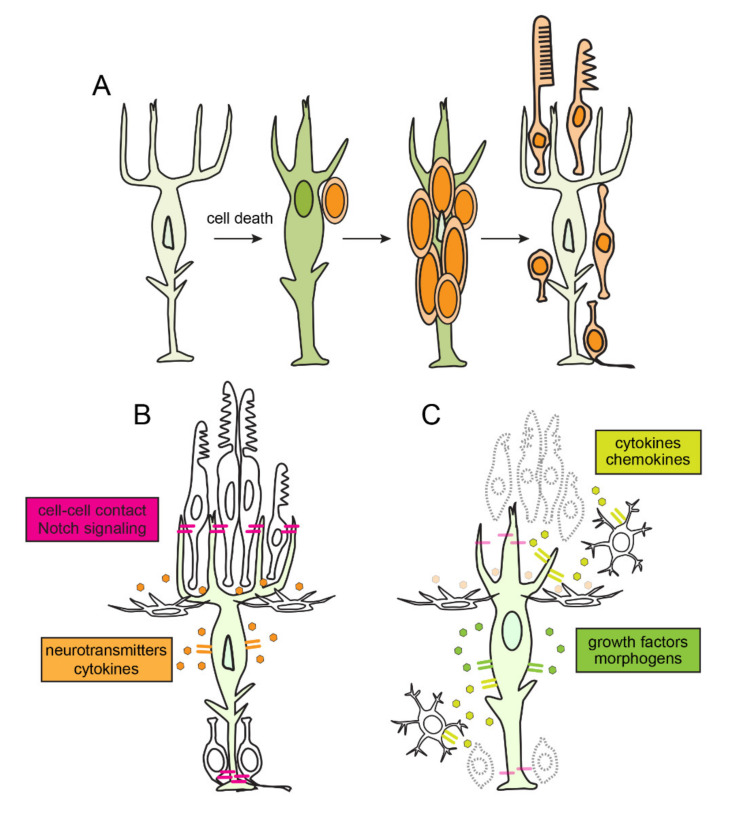
Retinal regeneration, regulation of quiescence and reprogramming of Müller glia in zebrafish. (**A**) Time course of retinal regeneration. Following cell death, Müller glia dedifferentiate and reprogram into a stem cell-like state. Müller glia then undergo interkinetic nuclear migration an asymmetric self-renewing division, which produces a multipotent retinal progenitor. Müller glia-derived progenitors then proliferate rapidly, migrate and differentiate into the ablated retinal neurons. (**B**) In an unlesioned retina, cell-cell contact mediated signaling, neurotransmitter dynamics, and autocrine signaling mechanisms maintain Müller glia in a quiescent state. (**C**) In response to cell death, soluble factors, cytokines, and chemokines, secreted from dying neurons and activated microglia induce reprograming of Müller glia. Müller glia themselves secrete growth factors and morphogens which promote reprogramming in autocrine manner.

**Figure 2 cells-10-00783-f002:**
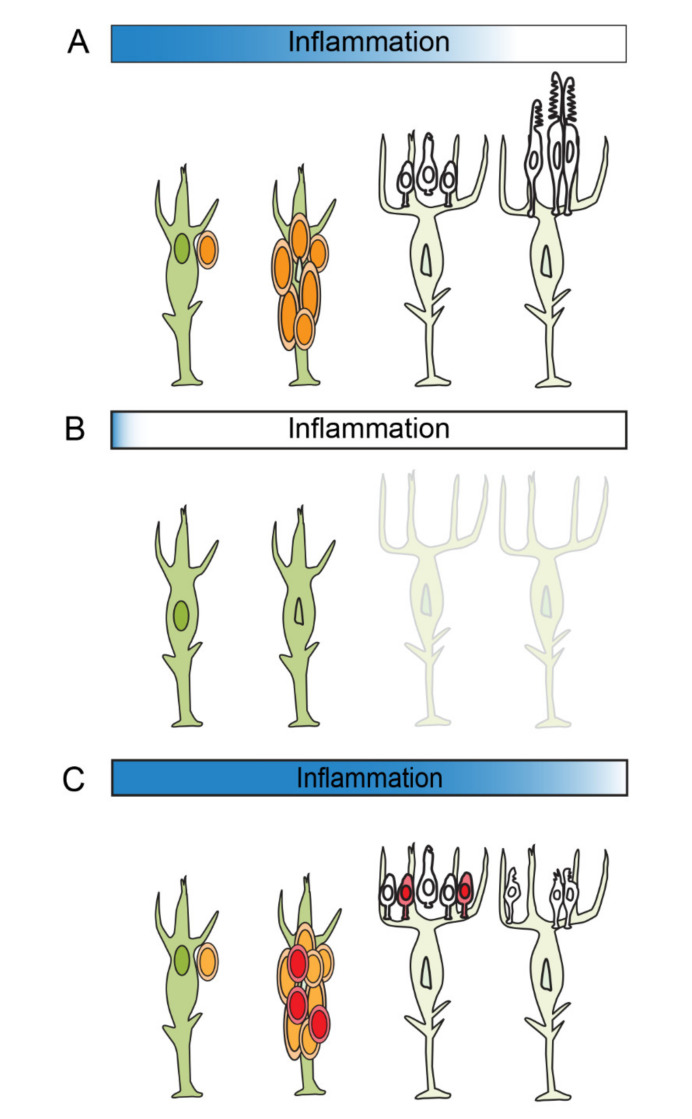
Model of how inflammation regulates photoreceptor regeneration in zebrafish. (**A**) Following cell death, acute inflammation stimulates reprogramming and cell division among Müller glia and proliferation of Müller glia-derive progenitors. As inflammation resolves, Müller glia-derived progenitors exit the cell cycle and differentiate into photoreceptors. (**B**) Suppression of acute inflammation results in the failure of Müller glia to proliferate. (**C**) If acute inflammation is prolonged, Müller glia-derived progenitors undergo extra rounds of cell division. If resolution of acute inflammation fails, this compromises the maturation and survival of photoreceptors.

## Data Availability

Not applicable.

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
