# Peer review of "Inflammation Regulates the Multi-Step Process of Retinal Regeneration in Zebrafish"

_cells, 2021, doi:10.3390/cells10040783_

Round 1

Reviewer 1 Report

The manuscript from Nagashima and Hitchcock is a review focusing on the contribution of inflammation to the process of retinal regeneration in the fish retina. This is very timely, considering the recent raise of interest in this topic. The manuscript is well written and organized, very clear and well referenced. I only have minor comments.

  1. Line 246, the authors wrote “It will be important also to identify the molecular mechanisms that constrain proliferation of Müller glia and their subsequent rapid return to quiescent state.”. They could mention here Nfi factors, which were shown to restore a glial quiescent state (Hoang et al., Science 2020).

  1. Line 303, the authors wrote “Together the concentration and/or perdurance of inflammatory molecules released into the extracellular environment by microglia and Müller glia regulate cell cycle kinetics of the Müller glia-derived progenitors, thereby determining the number that are produced.”. It is not clear whether this is a hypothesis. If not, the reference showing an effect on cell cycle kinetics should be cited. Is the number of rounds of divisions also a key parameter to control the number of produced cells?

  1. The schematic in Figure 1B suggests an interaction between photoreceptors and Müller cells through Notch signaling. Are there really some data suggesting that Notch signaling in Müller cells is regulated by photoreceptors?

  1. Some minor edits:
  1. line 157: It remains to be determined regeneration-specific molecules or mechanisms are involved in clearing 158 apoptotic cells
  2. line 165: Müller glia also display features of typical of astrocytes,
  3. line 184: mulit-phase response.
  4. Line 200 : and Müller glia themselves, Müller glia activate
  5. Line 230: photoreceptors. (Although Müller glia fail to progress through 230 the cell cycle, ablated rod photoreceptors are replenished, however, from the extant pool 231 of rod precursors.)

Reviewer 2 Report

This is a timely and interesting review discussing a significant topic. It is become clear that inflammation is a critical driver and regulator of the retinal regenerative response in zebrafish. There are some aspects the manuscript that should be addressed before publication:

Part 1: inflammation—the authors do not quite connect triggers of inflammation to the induction of cytokines, which is a downstream event. In the first few lines, they fail to indicate the nature of and how certain molecules act as triggers (for example PAMPs which are microbial in origin and DAMPs which are released from dying cells). It is important that they clearly convey that cytokines are not themselves triggers but instead are the downstream products that can then further play into cellular responses in the tissues. Also, in this section the authors could be clearer in that expression of cytokines/cytokine receptors are not limited to immune cells/leukocytes and that the expression of cytokine receptors can vary considerably by cell type and temporally during varying contexts. Further, most cytokine receptors trigger a signaling cascade that induces JAK/STAT pathways. The NFkB and AP-1 activation are more directly downstream of direct PAMP/DAMP recognition by pattern recognition receptors expressed on various cell types, especially leukocytes.

In addition, the authors should better acknowledge that inflammatory signals include pathways/molecules beyond cytokines. These include the complement cascade, leukotrienes, arachidonic acid pathway, histamines, etc.

Part 3: activation of microglia

-many of the cytokines noted in lines 47-48 were also found in the bulk RNA-seq done by Mitchell et al. 2019

Part 4: Quiescence, Reprogramming and Proliferation in Müller Glia

When discussing TGFb it could be worth connecting this to work showing that TGFb is important for establishing microglia phenotype

Line 214: should be specific about which Jak/STAT are involved here

Part 5: Inflammation and the Amplification of Müller Glia-Derived Progenitors

In regards to the idea of levels of cell death predicting Muller glia response, it is worth considering that the nature of the inflammation triggering molecule(s) in most contexts is not known. Importantly, it is worth revisiting that these different injury paradigms likely induce different forms of cell death and therefore the types of DAMPs that are released from the dying cells. For example apoptosis vs necrosis and the release of signals or DAMPs from the dying cells is significantly different. Along those lines, different DAMP molecules can trigger varying responses at different levels/intensities. Different DAMPs can have varying effects on innate immune cell (and other cell type) inflammatory responses. In addition, the clearance time of these dying cells could be important as it allows for temporal control of the availability of triggering molecules. This also circles back to the idea of removing the stimulus of inflammation.

Line 339-344: The authors should consider and briefly discuss the literature regarding effects of apoptotic cell recognition and engulfment on macrophage phenotype.

Part 7—Summary and future

It is worth considering that the molecular nature of the inflammatory signals could vary significantly between zfish and mammals, and this is yet to be determined.

In discussing the findings regarding zebrafish Tregs—perhaps the microglia/innate response drives their influx?

Round 2

Reviewer 2 Report

All revisions are sufficient.